# Gross and Micro-Anatomical Study of the Cavernous Segment of the Abducens Nerve and Its Relationships to Internal Carotid Plexus: Application to Skull Base Surgery

**DOI:** 10.3390/brainsci11050649

**Published:** 2021-05-16

**Authors:** Grzegorz Wysiadecki, Maciej Radek, R. Shane Tubbs, Joe Iwanaga, Jerzy Walocha, Piotr Brzeziński, Michał Polguj

**Affiliations:** 1Department of Normal and Clinical Anatomy, Chair of Anatomy and Histology, Medical University of Lodz, ul. Żeligowskiego 7/9, 90-752 Łódź, Poland; michal.polguj@umed.lodz.pl; 2Department of Neurosurgery, Spine and Peripheral Nerve Surgery, Medical University of Lodz, University Hospital WAM-CSW, 90-549 Łódź, Poland; maciej.radek@umed.lodz.pl; 3Department of Neurosurgery, Tulane Center for Clinical Neurosciences, Tulane University School of Medicine, New Orleans, LA 70112, USA; shane.tubbs@icloud.com (R.S.T.); iwanagajoeca@gmail.com (J.I.); 4Department of Neurosurgery and Ochsner Neuroscience Institute, Ochsner Health System, New Orleans, LA 70433, USA; 5Department of Neurology, Tulane Center for Clinical Neurosciences, Tulane University School of Medicine, New Orleans, LA 70112, USA; 6Department of Anatomical Sciences, St. George’s University, Grenada FZ 818, West Indies; 7Department of Surgery, Tulane University School of Medicine, New Orleans, LA 70112, USA; 8Department of Anatomy, Kurume University School of Medicine, 67 Asahi-machi, Kurume, Fukuoka 830-0011, Japan; 9Department of Anatomy, Jagiellonian University Medical College, 33-332 Kraków, Poland; jwalocha@cm-uj.krakow.pl; 10Department of Histology and Embryology, Chair of Anatomy and Histology, Medical University of Lodz, 90-752 Łódź, Poland; piotr.brzezinski@umed.lodz.pl

**Keywords:** abducens nerve, cavernous sinus, internal carotid plexus, dura mater/anatomy, microsurgery, skull base/anatomy

## Abstract

The present study aims to provide detailed observations on the cavernous segment of the abducens nerve (AN), emphasizing anatomical variations and the relationships between the nerve and the internal carotid plexus. A total of 60 sides underwent gross-anatomical study. Five specimens of the AN were stained using Sihler’s method. An additional five specimens were subjected to histological examination. Four types of AN course were observed: a single nerve along its entire course, duplication of the nerve, division into separate rootlets at the point of contact with the cavernous part of the internal carotid artery (ICA), and early-branching before entering the orbit. Due to the relationships between the ICA and internal carotid plexus, the cavernous segment of the AN can be subdivided into a carotid portion located at the point of contact with the posterior vertical segment of the cavernous ICA and a prefissural portion. The carotid portion of the cavernous AN segment is a place of angulation, where the nerve always directly adheres to the ICA. The prefissural portion of the AN, in turn, is the primary site of fiber exchange between the internal carotid plexus and either the AN or the lateral wall of the cavernous sinus.

## 1. Introduction

Advanced knowledge of the cavernous sinus (CS) anatomy is of great clinical importance, especially in neurosurgery [1,2,3]. Beginning with Parkinson’s early pioneering work [4], the CS gradually ceased to be “surgical no man’s land”. However, operative procedures in this area remain challenging due to the complex anatomical relationships and difficulty of the surgical approaches. As Isolan et al. [5,6] stressed, the CS is characterized by a unique meningeal lining and concentration of essential vascular and neural elements. The cavernous portion (C4 segment) of the internal carotid artery usually has an S-shaped course within the CS [7]. The ICA is accompanied by a network of sympathetic nerves known as the internal carotid plexus (ICP) [7,8,9,10]. The abducens nerve (AN) runs between the tortuous cavernous carotid siphon and the lateral wall of the CS; The oculomotor, trochlear, and ophthalmic nerves, in turn, run in the lateral wall of the CS without being in direct contact with the venous blood [5,6,11,12,13].

A thorough understanding of the topographical anatomy of the neural structures related to the CS can explain clinical symptoms in disease processes involving this area and is helpful during neurosurgical procedures. The cavernous segment of the AN can be involved in numerous pathologies of neoplastic, vascular, and inflammatory origin. Due to its unique spatial relationships, the cavernous segment of the AN is susceptible to iatrogenic injury during endovascular interventions within the ICA and skull base surgery for CS lesions [14]. The AN also intersects with the bundles of sympathetic fibers derived from the ICP, some of which run toward the nerve or join the cranial nerves located in the lateral wall of the CS [8,15]. Observations of the arrangement of the ICP, its relationships to the cavernous portion of the AN and ICA, as well as communications with the lateral wall of the CS, may provide structural background for understanding the sympathetic nerve supply of the orbit and innervation of the cerebral vasculature [8,15,16].

The present study aims to provide detailed observations on the AN cavernous segment, emphasizing anatomical variations and the relationship between the nerve and the ICP. To supplement gross-anatomical investigation, histological techniques were used to show detailed nerve relationships under magnification. Additionally, a specialized whole mount nerve staining technique was used.

## 2. Materials and Methods

### 2.1. Gross Anatomical Study and Measurements

The study was conducted according to the Declaration of Helsinki guidelines and approved by the Bioethics Committee of the Medical University of Lodz (protocol code: RNN/518/14KB, with further amendment KE/322/21). The classification of the cavernous segment of the AN was based on the findings of Iaconetta et al. [11]. Selected terms applied to describe sympathetic fiber bundles in the CS were adapted from van Overbeeke et al. [15]. 

Ten adult cadaveric heads and 40 sagittal head sections fixed in 10% formalin were used for the gross-anatomical study. The total amount of specimens (also referred to as sides or cases in this report) subjected to gross-anatomical study was 60. Evidence of past trauma or surgical interventions was not found in any of the specimens. Anatomical dissection was conducted at 2.5× magnification using HEINE^®^ HR 2.5 X High-Resolution Binocular Loupe (HEINE Optotechnik GmbH & Co. KG, Herrsching, Germany). The incision was made along the interclinoid dural fold. The fold’s attachment was released from the anterior and posterior clinoid processes, and the lateral wall was carefully reflected and removed. During the study, anatomical variations of the AN were assessed, along with the nerve’s relation to the ICP. The research was supplemented with morphometric measurements using a digimatic digital caliper (Mitutoyo Company, Kawasaki-shi, Kanagawa, Japan). The length of the cavernous segment of the AN was taken. The petrous apex and upper border of the petrolingual ligament were used as landmarks. The distances were measured between the upper border of the petrolingual ligament and AN (at the level where the nerve crossed the posterior vertical segment of the cavernous part of the ICA), as well as between the petrous apex and the point where the anterior bundle of sympathetic fibers crossed the AN.

### 2.2. Sihler’s Staining

A whole-mount nerve staining technique (Sihler’s stain) was used to visualize the course of the cavernous segment of the AN. The staining procedure was conducted according to the protocol described by Mu and Sanders [17,18] and Won et al. [19]. As Mu and Sanders [17] stressed, Sihler’s stain clears soft tissue while counterstaining all nerves. Five specimens of the AN, including the cavernous part of the ICA and ICP, were harvested en bloc, rinsed under running tap water, and subjected to a series of subsequent staining steps. These steps involved maceration in potassium hydroxide (with a slight admixture of hydrogen peroxide), decalcification in acetic acid, staining in Ehrlich’s hematoxylin, destaining in acetic acid (with chloral hydrate), and clearing in glycerin.

A Canon EOS 5D Mark II digital camera was used to record the results. The specimens were placed on a negatoscope and photographed against a light background. To obtain optimal reflection of small anatomical details, without the sticking of tiny nerves, we introduced our original solution; examined samples were immersed in a Petri dish filled with glycerin and then photographed. This allowed the anatomical details to be accurately reproduced, and the adhesion of very fine nerves was avoided.

### 2.3. Histological Examination

An additional five specimens were subjected to histological examination. The CS was harvested en bloc with the lateral wall, AN, cavernous part of the ICA, and ICP for the microanatomical study. Samples obtained this way were cleaned by rinsing with 0.9% sodium chloride solution and fixed in 4% buffered formalin. In two specimens, the lateral wall of the CS was removed. In these samples, a specimen was embedded in paraffin in order to have histological sections made parallel to the nerve’s long axis (in the sagittal plane). The other two specimens were embedded in paraffin and sectioned in the frontal (coronal) plane. These specimens included the lateral wall of the CS. One specimen involved the area where some potential communications between the ICP and lateral wall of the CS were macroscopically observed. Mallory’s trichrome stain was used as a reference method to visualize connective tissues on histological sections. The Gordon and Sweet’s silver staining method was used to visualize reticular fibers, which are components of peripheral nerve fibers. This method was selected as it gives greater contrast and sharpness against the background. Histological slices were assessed using OPTA-TECH MB 200 Series Biological Microscope (OPTA-TECH, Warsaw, Poland) with OPTA-TECH HDMI CAM Microscope camera with HDMI CAM embedded software.

## 3. Results

### 3.1. Gross Anatomical Observations

The cavernous segment of the AN traveled between the petrous apex and the superior orbital fissure (Figure 1). The mean length of this segment was 26.9 mm (median = 26.7 mm, minimum = 19.2 mm, maximum = 32.7 mm, SD = 3.3). It crossed the proximal part of the cavernous part of the ICA, a few millimeters (mean = 4 mm, median = 4.3 mm, minimum = 1.3 mm, maximum = 6.5 mm, SD = 1.4) over the superior border of the petrolingual ligament. At the crossing point with the posterior vertical part of the cavernous ICA, the AN angulation was observed in all specimens. At this point, the nerve’s epineurium was loosely fixed to adventitia covering the lateral wall of the posterior vertical part of the cavernous ICA. The AN inferior aspect was joined by a well-developed anterior bundle of the sympathetic fibers derived from the ICP (Figure 1 and Figure 2), on average 9.7 mm from the petrous apex (median = 8.9 mm, minimum = 7.1 mm, maximum = 17.3 mm, SD = 2.2). This characteristic point was a segregation place (a kind of “marshaling yard”) for the sympathetic fibers of the ICP. Some fibers continued their course along the ICA; some of them accompanied the AN and then joined the lateral wall of the CS, while some fibers continued with the AN anteriorly.

### 3.2. Anatomical Variations

Four arrangements of AN course were observed on the examined sample. In most cases (38/60; 63.3%), a single AN trunk coursed within the CS as far as the superior orbital fissure (Figure 1a and Figure 2a). In 15 cases (15/60; 25%), division of the AN into separate rootlets (“pseudobranching”) was observed at the point of contact with the posterior vertical segment of the cavernous part of the ICA (Figure 1b and Figure 2c). The number of separate fascicles varied from two to four in the examined sample (mean = 2.6; median = 2; SD = 0.76). The distance at which this branching was observed varied from 7.1 mm to 12.9 mm (mean = 9.4 mm; median = 9.2 mm; SD = 1.7 mm). On four sides (4/60; 6.7%), the AN was duplicated (Figure 3). On these sides, both AN nerve trunks merged within the CS. In another three cases (3/60; 5%), the “early branching” of the AN was observed in the CS (Figure 2b). On these sides, the AN divided into two separate branches before reaching the orbit; those branches continued a parallel course through the superior orbital fissure. The frequency of the AN variations found in our study is presented in Table 1.

### 3.3. Sihler’s Staining

Five specimens harvested from right sides were stained using Sihler’s technique. For comparison, each type of anatomical variation was examined with this method; we used one sample of AN duplication, one sample showing AN branching within the CS, one specimen with AN early division into branches, and two specimens with typical single AN trunk. Although easy to separate from the ICA, the nerve retained attached to the artery’s lateral wall during all steps of Sihler’s staining in four out of five specimens. The method allowed for better visualization of both the main AN trunk and some delicate nerve bundles accompanying the nerve. On the specimens stained using Sihler’s technique, the well-developed anterior bundle of the sympathetic fibers, emerging from the ICP surrounding the anterior aspect of the petrous ICA, joined the inferior aspect of the nerve (Figure 2 and Figure 3). Small nerve fiber bundles (fascicles) derived from the ICP located along ICA posterior genu also entered the nerve’s medial aspect (Figure 2). Detailed examination of specimens prepared using a whole-mount nerve staining technique confirmed the presence of the fine nerve fiber bundles that continued with the AN anteriorly (Figure 3) toward the superior orbital fissure. However, careful examination of the duplicated AN stained with Sihler’s method revealed that the majority of fibers derived from the ICP were related to (accompanied) the most medial trunk of the nerve, which directly adhered to the wall of the ICA (Figure 3).

### 3.4. Histological Examination

The most typical single AN was found in the transverse CS sections. The AN was found on the medial side of the lateral wall of the CS, just at the level of the trigeminal nerve’s ophthalmic division (Figure 4 and Figure 5). At the level of the horizontal portion of the cavernous part of the ICA, the AN trunk was composed of few fascicles surrounded by common epineurium composed of dense irregular connective tissue (Figure 4). Some fine nerve fascicles derived from the ICP were observed under cover of the AN epineurium and in the nerve trunk’s close vicinity (Figure 5). Two layers in the lateral wall of the CS were marked; the external dural layer was composed of dense irregular connective tissue forming thicker external covering, while the deep layer (reticular membrane) was seen as irregular collagen fibers surrounding nerves running in the wall. The reticular membrane contained numerous fascicles of the ophthalmic nerve, the trochlear, and the oculomotor nerve. Some communications between the ICP and lateral wall of the CS were observed. The presence of those fine nerve bundles was confirmed histologically (Figure 6).

Branching into few separate fascicles (“pseudobranching”) of the AN was visualized in one sagittally sectioned specimen (Figure 7 and Figure 8). The AN division area was characterized by the nerve’s plexiform appearance with few interconnections between the nerve’s fascicles (Figure 7 and Figure 8). On both specimens sectioned in the sagittal plane, the anterior bundle of the ICP was identified inferior to the AN (Figure 8). The delicate bundles derived from the ICP sent communications with the AN. Fine nerve bundles were also identified in the nerve’s surroundings. Small blood vessels were also found in this area.

### 3.5. Proposal of AN Cavernous Segment Subdivision

Due to the relationships to the ICA and ICP, we suggest that the cavernous segment of the AN could be subdivided into the carotid (at the point of contact with the posterior vertical part of the cavernous segment of the ICA) and prefissural portions (between the anterior aspect of vertical ICA and the superior orbital fissure; see Figure 9). The carotid portion of the cavernous AN segment shows angulation, where the nerve always adheres directly to the ICA. The prefissural portion of the AN, in turn, is the primary site of fiber exchange between the ICP and either AN or the lateral wall of the CS. This subsegment is characterized by the presence of small sympathetic fibers accompanying the AN into the orbit.

## 4. Discussion

Iaconetta et al. [11] divided the AN into five segments due to the nerve’s specific course and anatomical relationships; Out of those segments, three are intracranial (cisternal, gulfar, and cavernous), and two are orbital (fissural and intraconal). According to the various authors, the specific spatial relations determining the division into five segments are apparent using microsurgical transcranial and endoscopic endonasal approaches [11]. Zhang et al. [20] proposed a similar classification, with the difference that the cavernous segment of the AN was referred to as “ICA segment.” Li et al. also divided the AN into five segments: the cisternal segment, Dorello’s canal segment, CS segment, superior orbital fissure segment, and intraorbital segment [21]. Based on AN relation to the ICA and sympathetic fibers derived from the ICP, we suggest refining the classification and subdivision of the nerve’s cavernous segment into two portions (subsegments), i.e., carotid (at this level, angulation of the nerve takes place as the ICA) and prefissural, which is the main area of intersection with small nerve bundles derived from the ICP. Such a subdivision seems to be justified due to the unique neurovascular topographical relations within each cavernous AN subsegment (Figure 9) and the use of such a classification with modern microsurgery.

Few anatomical variations of the cavernous segment of the AN might be observed (Figure 10). According to the classical textbook description, the single trunk of the AN courses within the CS (Figure 10a). These relationships, however, are more diverse. The phenomenon of AN “pseudobranching”, i.e., separation of the nerve’s cavernous segment into several fascicles at the crossing site with the posterior vertical portion of the cavernous part of the ICA (Figure 10b,c), is a common anatomical variant, with a reported frequency ranging from 20 to 37.5% [22,23,24,25]. In the presented study, we found this variant in 25% of cases, with two to four separate fascicles at the area where the AN crossed the vertical segment of the cavernous part of the ICA. Harris and Rhoton [22] observed up to five separate rootlets at the contact site between the AN and ICA. Ozer et al. [23] first suggested that this AN branching pattern in the CS might be called “pseudobranching.” Such a term seems to be justified as the individual fascicles merge into a common trunk after a short course ranging from 7.1 mm to 12.9 mm in this study, from 5 mm to 12 mm according to Zhang et al. [20], and from 4.5 mm to 9.5 mm (mean: 8.4 mm, SD: 1.8 mm) according to another report of Wysiadecki et al. [25]. This indicates that “pseudobranching” of the AN can take place in various ways and at different lengths; however, when present, pseudobranching always involves the nerve’s point of contact with the posterior vertical segment of the cavernous ICA. We also demonstrated interconnections between the AN rootlets at the pseudobranching site in a histological examination of this anatomical variant (Figure 7 and Figure 8).

Duplication is another anatomical variation of the AN (Figure 10d–f) [20,23,25,26,27,28,29,30,31]. It should be stressed that AN duplication typically involves at least two segments of this nerve (in contrast to the nerve’s “pseudobranching” within the CS) [23,25]. Two main patterns of AN duplication have been reported [20,23,25]. The AN may leave the brainstem (pontomedullary sulcus) as two separate trunks. In such cases, duplication involves the cisternal, gulfar, and (commonly) cavernous segments of the nerve (Figure 10e,f). Duplication involving only the gulfar and cavernous segments can also be observed (Figure 10d). In the majority of cases, both trunks of the duplicated AN merge within the CS [20,23,25,26,27,28,29]. In our series, AN duplication (two main types described above) was observed in 6.7% of cases, and this frequency is less than that found in the studies of Nathan et al. (13.5%) [26], Ozveren et al. (15%) [27] and Zhang et al. (10%) [20]. Two cases are described in reports in the literature in which duplication of the AN involved the nerve’s entire intracranial course [30,31]. Another rare anatomical variant involves cases in which the AN is divided into two terminal branches within the CS (Figure 10g). We found such a variation in three out of 60 sides (5%). This variant was also reported by Ozer et al. [23] with a frequency of 7.5%. One recent study regarding the orbital (intraconal) segment of the AN reported that the AN could be divided into terminal branches before entering the orbit [32]. Such early division can be explained by an arrangement of the AN terminal sub-branches; Two main groups of the AN sub-branches (superior and inferior) can be distinguished [31,32]. Duplication of the AN can be considered a normal variant and this might be seen on MRI [33]. Preoperative identification of duplication of the AN, along with recognition of neighboring surgical landmarks, might reduce the risk of iatrogenic nerve injury, especially during skull base surgeries, endovascular procedures, or other interventions involving the CS [11,23,27].

Within the CS, unique relationships between the ICP and AN are observed. Overbeeke et al. [15], in their study of the anatomy of the sympathetic pathways in the CS, found a systematic arrangement of sympathetic pathways in the CS. One group of authors found, “A clearly visible bundle of sympathetic fibers was seen in all specimens to run in an anterior direction from the petrous carotid artery and merge with the AN on its inferior side” [15]. That bundle was referred to as the anterior bundle of sympathetic fibers; In most cases, the bundle’s fibers merged into the sheath of the AN [15]. This observation gained confirmation from the histological examination conducted in our study—fine nerve fascicles derived from the ICP were found under the AN’s epineurium. Similar observations were described by Johnston and Parkinson [34]. Overbeeke et al. [15], with histology, confirmed the fusion of sympathetic and motor fibers in the AN. The anterior bundle of sympathetic fibers was described by Monro [35] as early as 1746 and termed a “reflected branch of the sixth cranial nerve” due to its curved course. Meckel [36] discovered that this branch was derived from the ICP. The anterior bundle of sympathetic fibers was identified in all specimens in the present study (Figure 1, Figure 2 and Figure 3 and Figure 7, Figure 8 and Figure 9). Mariniello et al. [37] conducted a microanatomical study to trace sympathetic fibers inside the CS. These authors provided detailed information on the course of the sympathetic fibers and duplicated AN and said, “In two cases (3.2%) a double trunk of AN directly arising from the pons was present: in these specimens, the sympathetic fibers followed only the larger of the two branches” [37]. This finding is similar to our observations. Bleys et al. [8] examined the distribution of sympathetic fibers within the CS using whole-mount preparations and a sensitive acetylcholinesterase method. These authors introduced the term “lateral sellar plexus” and concluded that the plexus is composed of the main part (referred to as “the lateral sellar plexus proper”) located around the AN and medial to the ophthalmic nerve, with an inferior extension medial to the ophthalmic nerve, and a lateral extension just underneath the outermost layer of the lateral wall of the CS [8]. Knowledge of the surgical anatomy of the dural walls of the CS is crucial regarding new minimally invasive surgical techniques [12,13]. Thorough knowledge of the detailed topographic anatomy of the lateral wall of the CS, including relationships to the AN and ICA, as well as communications with the ICP, is relevant to the development of modern neurosurgical approaches.

The AN relationship to the cavernous part of the ICA and ICP also has clinical implications. As Iwanaga et al. [9] concluded, neurosurgeons who operate in and around the CS should be aware of such relations as unwanted tension on the cavernous part of the ICA or AN during dissection can result in Horner’s syndrome, miosis, increased accommodation, or ocular hypotony. AN palsy can also result from an aneurysm or arterial dissection of the cavernous part of the ICA as those conditions can cause direct compression to the nerve or interruption of its blood supply [38]. Isolated AN palsy due to ICA aneurysms are rare clinical issues. However, as Kim and Park [38] suggest, clinicians should consider ICA aneurysms when patients present isolated AN palsy. Mendez Roberts and Grimes [39] emphasize, in turn, that the proximity of cranial nerves II to VI to the course of the ICA makes these nerves susceptible to damage from an ICA aneurysms. According to Elder et al. [40], the etiologies of cavernous AN segment palsy include: cavernous part of the ICA aneurysm, neoplastic infiltration, idiopathic inflammation (Tolosa-Hunt syndrome), infection, and cavernous-carotid fistula. Thus, a thorough knowledge of topographical relations without the CS may contribute to diagnosing and treating disease processes that may involve this area. Newman [41] concludes that CS surgery results in transient worsening of cranial nerves III–VI function. Consequently, the ability to conduct successful surgeries within this area is based mainly on an improved understanding of the anatomy and pathology in the parasellar and petroclival regions combined with modern neuroimaging [10,14,42,43].

### Study Limitation

Since isolated specimens were used in the study, we could not assign age or sex to a specific specimen. Due to the arteries not being injected on the examined sample, the study was focused on nerves and did not involve branches of the inferolateral trunk and vascular supply of individual AN variants. Specific immunohistochemical techniques should be applied to evaluate the nerve fibers types running between the ICP and either the AN or the lateral wall of the cavernous sinus (especially to examine eventual presence of sensory fibers running from the trigeminal ganglion to the cavernous part of the ICA and cerebral vessels). 

We used gross and microanatomical techniques to provide a more in-depth understanding of complex relationships between the cavernous segment of the AN and adjacent structures. Knowledge of the presence and distribution of communications between the lateral CS wall and ICP may contribute to our improved understanding of the physiology in this region, e.g., innervation of the cerebral vessels or autonomic innervation of structures within the orbit [8,16,44].

## 5. Conclusions

Four types of AN course were observed in the present study: single nerve along its entire course, duplication of the nerve, pseudobranching (i.e., division into separate rootlets at the point of contact with the cavernous ICA), and early-branching before entering the superior orbital fissure. Due to the relationships to the ICA and ICP, the cavernous segment of the AN can be subdivided into a carotid portion located at the point of contact with the cavernous part of the ICA and the prefissural portion. The carotid portion of the cavernous AN segment is a place of angulation, where the nerve is always directly adherent to the ICA. The prefissural portion of the AN is the primary site of fiber exchange between the ICP and either the AN or the lateral wall of the CS.

## Figures and Tables

**Figure 1 brainsci-11-00649-f001:**
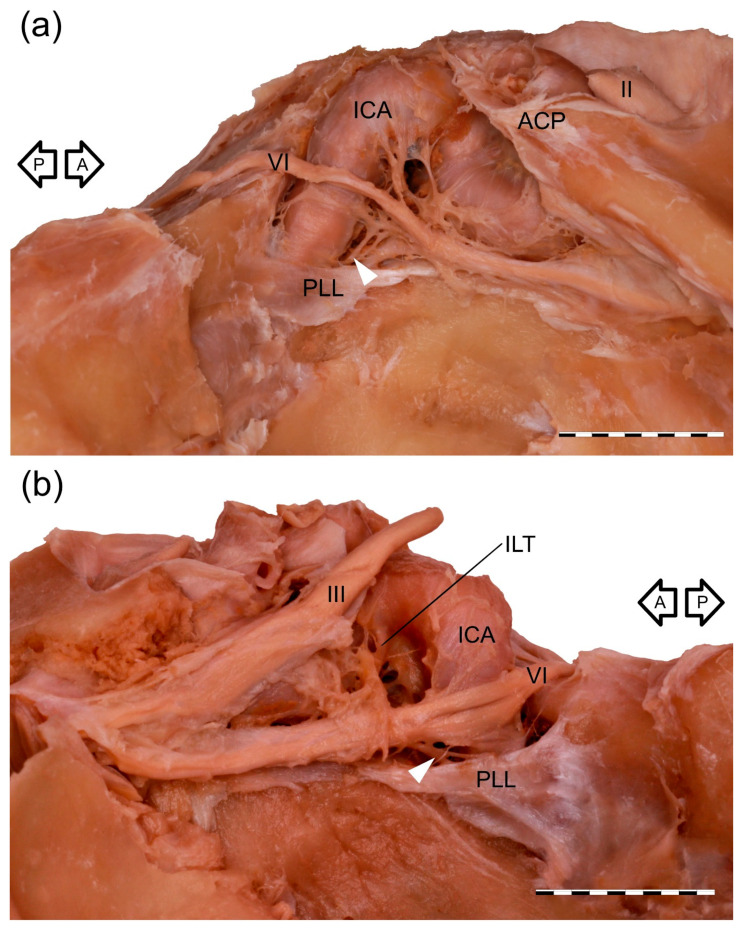
Anatomical variations of the AN cavernous segment. Note the different distances of the AN (VI) to the petrolingual ligament (PLL) superior border. White arrowhead marks an anterior bundle of the sympathetic nerve fibers joining the AN’s inferior aspect. (**a**) Wet anatomical specimen of the right CS. A single AN (VI) running within the CS. ACP—anterior clinoid process; II—optic nerve; III—oculomotor nerve. (**b**) Wet anatomical specimen of the left CS. Division of the AN (VI) into separate rootlets at the point of contact with the cavernous carotid artery’s vertical segment. III—oculomotor nerve; ILT—inferolateral trunk. Black arrows indicate anterior (A) and posterior (P) directions. Scale bar corresponds to 10 mm.

**Figure 2 brainsci-11-00649-f002:**
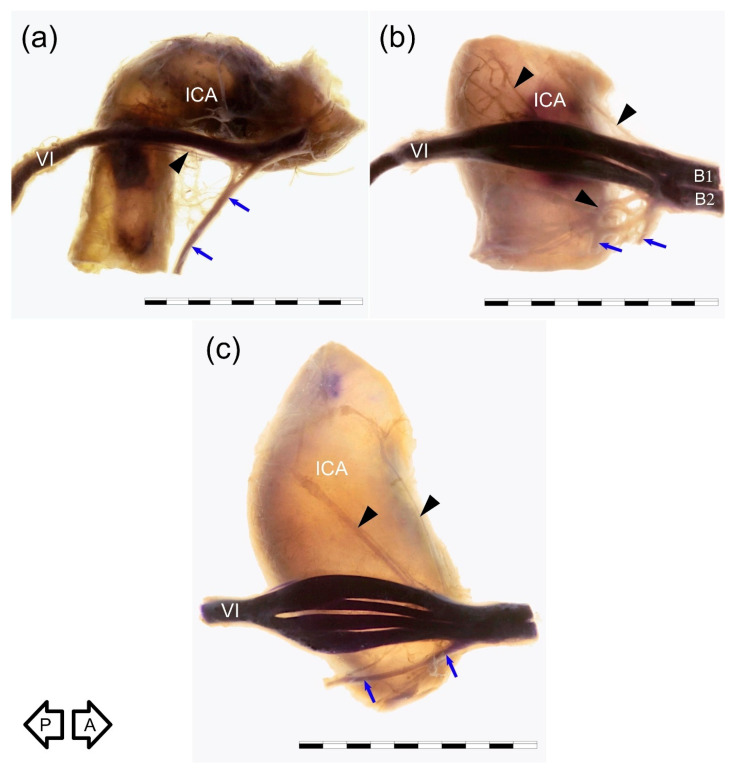
Abducens nerve (VI) visualized at the crossing point with the cavernous carotid artery (ICA). Lateral view of the specimens harvested from the right CS. Specimens were prepared using a whole-mount nerve staining technique (Sihler’s stain). Relation of the nerve to the ICP is also presented. Blue arrows mark a well-developed anterior bundle of the sympathetic nerve fibers joining the inferior aspect of the AN. Black arrows show tiny nerve bundles joining the medial aspect of the nerve. (**a**) Typical variant with single nerve. (**b**) “Early branching” of the AN. In this variant, the nerve was divided into two separate branches (B1 and B2) at the contact point with the cavernous part of the ICA posterior vertical segment’s lateral wall. In this case, separate AN branches continued to the superior orbital fissure. (**c**) Variant with AN divided into four separate rootlets at the point of contact with the posterior vertical segment of the cavernous ICA. Black arrows indicate anterior (A) and posterior (P) directions. Scale bar corresponds to 10 mm.

**Figure 3 brainsci-11-00649-f003:**
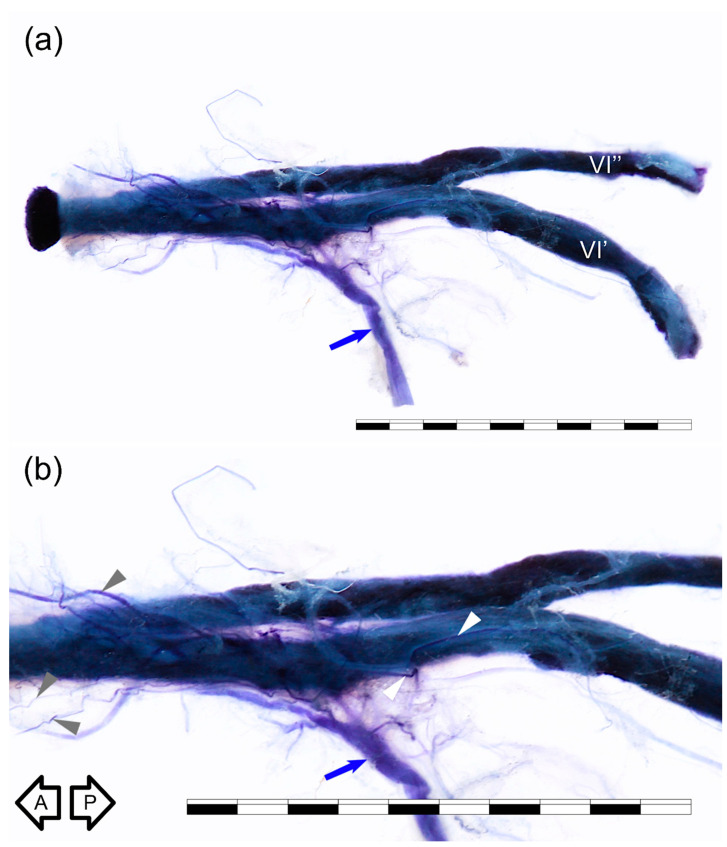
Duplication of the AN. Medial view of the right nerve. Sihler’s stain. Two nerve trunks (VI’ and VI’’) merge. (**a**) General view. (**b**) Photograph enlarged to visualize better nerve fiber bundles derived from the internal carotid plexus that accompany the AN. Blue arrow marks a well-developed anterior bundle of the sympathetic nerve fibers joining the inferior aspect of the nerve. Some of those fibers accompany the AN to join the cavernous sinus lateral wall (white arrowheads), while some of the fibers continue with the AN anteriorly (gray arrowheads). Black arrows indicate directions: A—anterior; P—posterior. Scale bar corresponds to 10 mm.

**Figure 4 brainsci-11-00649-f004:**
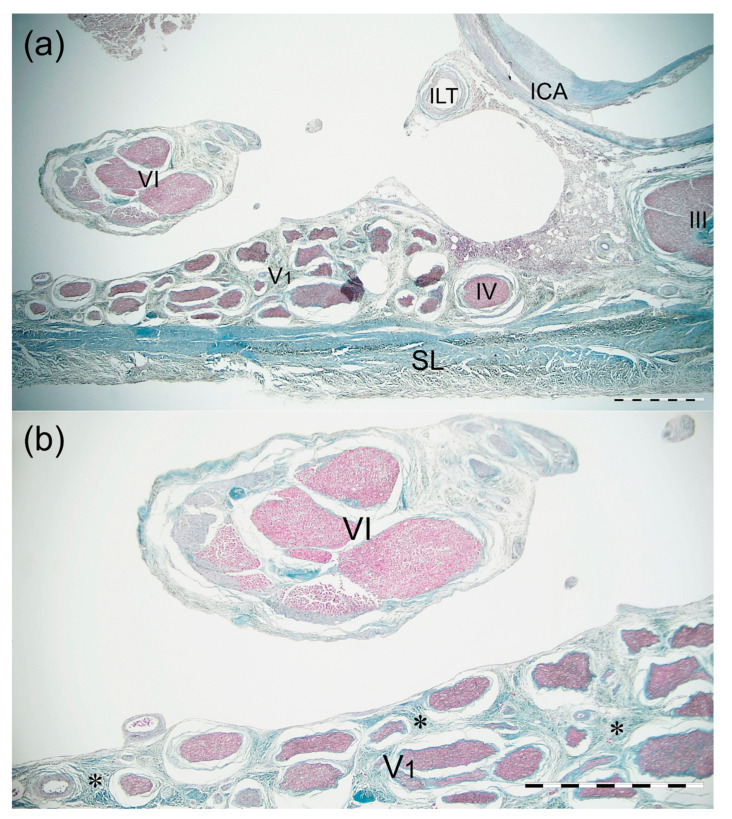
Cross-section of the AN, lateral wall of the CS, and the horizontal portion of the cavernous part of the ICA. Mallory’s trichrome stain. (**a**) Panoramic view of the specimen seen under 2 × objective lens. The AN (VI) runs on the medial side of the lateral wall of the CS, at the level of the trigeminal nerve’s ophthalmic division (V1). ICA—internal carotid artery; ILT—branch of the inferolateral trunk; SL—superficial layer of the lateral wall of the CS; III—oculomotor nerve; IV—trochlear nerve. (**b**) Magnification of the specimen seen in figure (**a**). 4 × objective lens. The single trunk of the AN (VI) contains a few fascicles surrounded by common epineurium composed of dense irregular connective tissue. Collagen fibers surround the fascicles of the trigeminal nerve’s ophthalmic division (V1), forming the deep layer (reticular membrane) of the lateral wall of the CS. Scale bar corresponds to 1 mm.

**Figure 5 brainsci-11-00649-f005:**
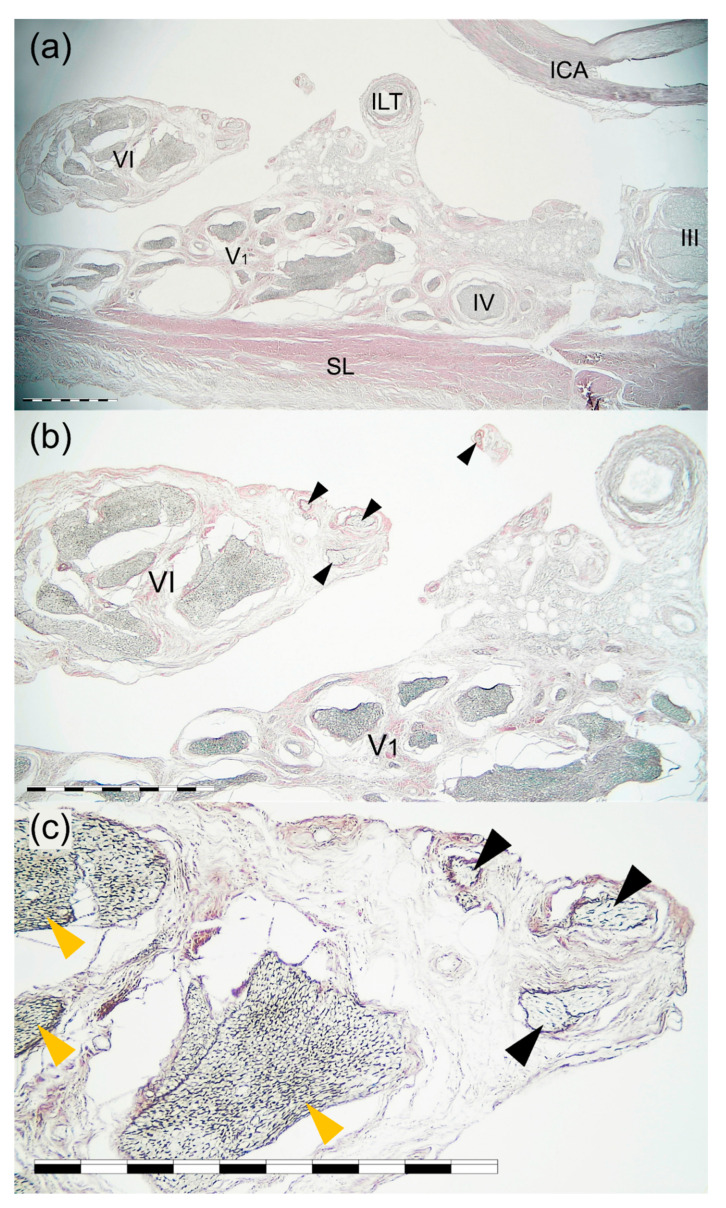
Cross-section of the AN, lateral wall of the cavernous sinus, and the horizontal portion of the cavernous part of the ICA. Silver staining according to Gordon and Sweet’s method. (**a**) Panoramic view of the specimen seen under 2 × objective lens. The AN (VI) runs on the medial side of the cavernous sinus lateral wall, at the level of the trigeminal nerve’s ophthalmic division (V1). ICA—internal carotid artery; ILT—branch of the inferolateral trunk; SL—superficial layer of the cavernous sinus lateral wall; III—oculomotor nerve; IV—trochlear nerve. (**b**) Magnification of the specimen seen in figure (**a**). 4 × objective lens. Fine nerve fascicles derived from the ICP (marked by black arrowheads) are visualized under cover of the AN (VI) epineurium and in the nerve’s close vicinity. (**c**) Magnification of (**b**) showing fine nerve fascicles (marked by black arrowheads) derived from the ICP and running under cover of the AN epineurium. Anatomical structures are seen under a 10 × objective lens. The visual difference in the content of fibers having myelin sheath can be observed between AN fascicles (marked by yellow arrowheads) and those derived from the ICP. Scale bar corresponds to 1 mm.

**Figure 6 brainsci-11-00649-f006:**
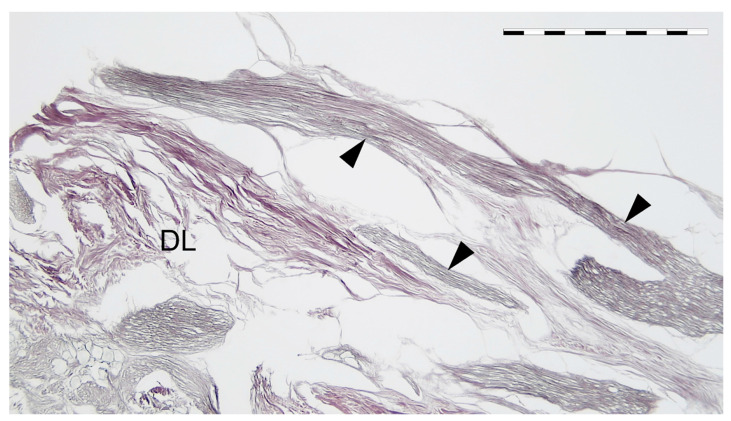
Communications between the ICP and the lateral wall of the CS (black arrowheads). Silver staining according to Gordon and Sweet’s method. 4 × objective. DL—deep layer (reticular membrane) of the lateral wall of the CS. Scale bar corresponds to 1 mm.

**Figure 7 brainsci-11-00649-f007:**
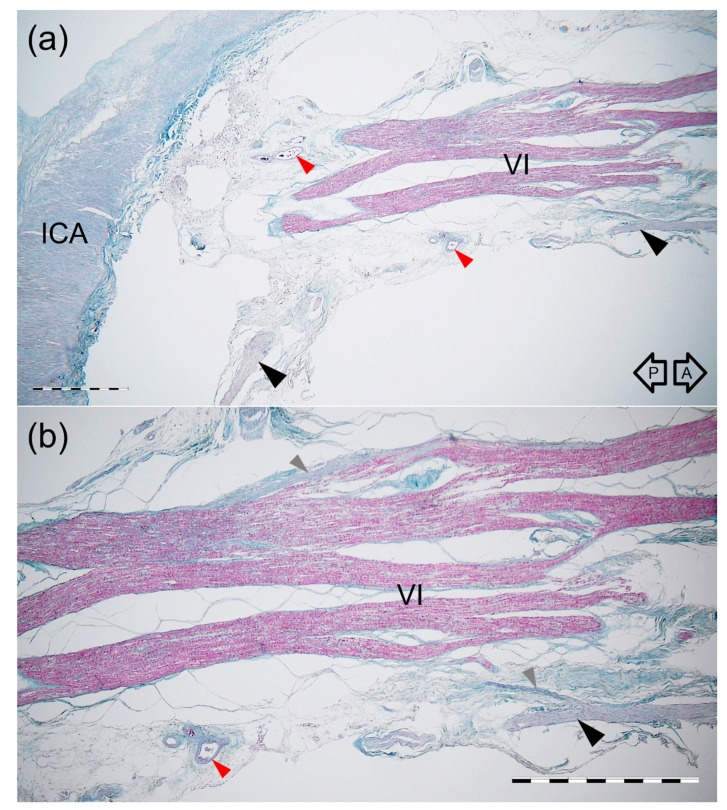
Histological specimen of the “pseudobranching” of the AN. Longitudinal section parallel to the nerve’s long axis. Mallory’s trichrome stain. (**a**) Panoramic view of the nerve seen under 2 × objective lens. The black arrowheads indicate the anterior bundle of sympathetic fibers which join the inferior surface of the AN. The red arrowheads show groups of small blood vessels. ICA—internal carotid artery. (**b**) Magnification of the specimen seen in figure (**a**). 4 × objective lens. The AN (VI) dividing into a few fascicles (“pseudobranching”) is characterized by a plexiform appearance with interconnections between them. The anterior bundle of the ICP (black arrowhead) is exposed on the AN’s inferior surface. The delicate bundles derived from the ICP sent communications to the AN (gray arrowheads). The red arrowhead shows a group of small blood vessels. The black arrows indicate anterior (A) and posterior (P) directions. Scale bar corresponds to 1 mm.

**Figure 8 brainsci-11-00649-f008:**
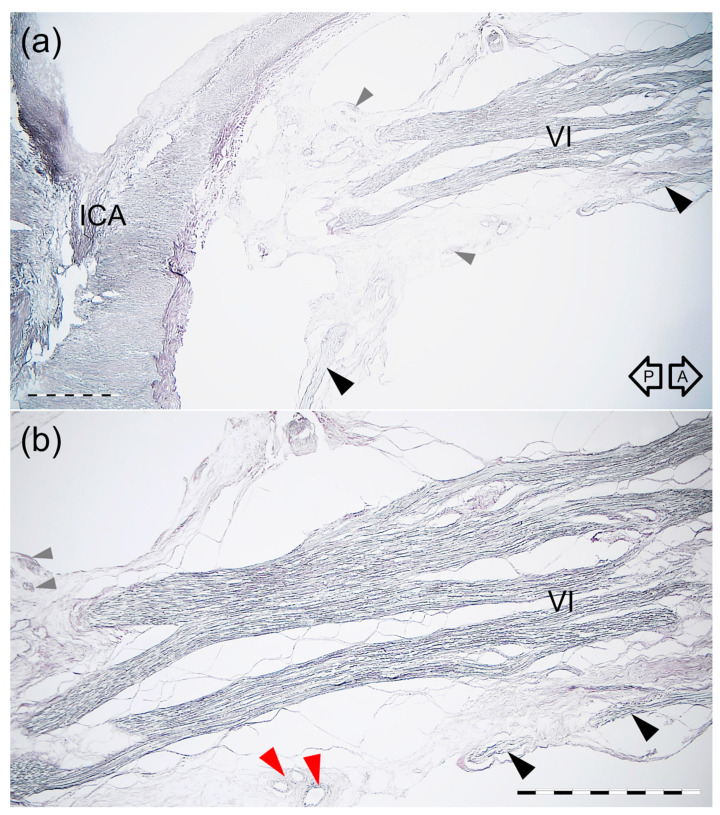
Histology of the “pseudobranching” of the AN. Longitudinal section parallel to the nerve’s long axis. Silver staining according to Gordon and Sweet’s method. (**a**) Panoramic view of the nerve seen using a 2 × objective lens. The black arrowheads indicate the anterior bundle of sympathetic fibers which join the inferior surface of the AN. Gray arrowheads indicate cross-sections of the delicate bundles derived from the ICP near the AN. ICA—internal carotid artery. (**b**) Magnification of the specimen presented in figure (**a**) seen under 4 × objective lens. The AN (VI) “pseudobranching” shows a plexiform appearance with interconnections between the fascicles. The anterior bundle of the ICP (black arrowhead) is exposed on the AN’s inferior surface. The red arrowhead shows a group of small blood vessels. The black arrows indicate anterior (A) and posterior (P) directions. Scale bar corresponds to 1 mm.

**Figure 9 brainsci-11-00649-f009:**
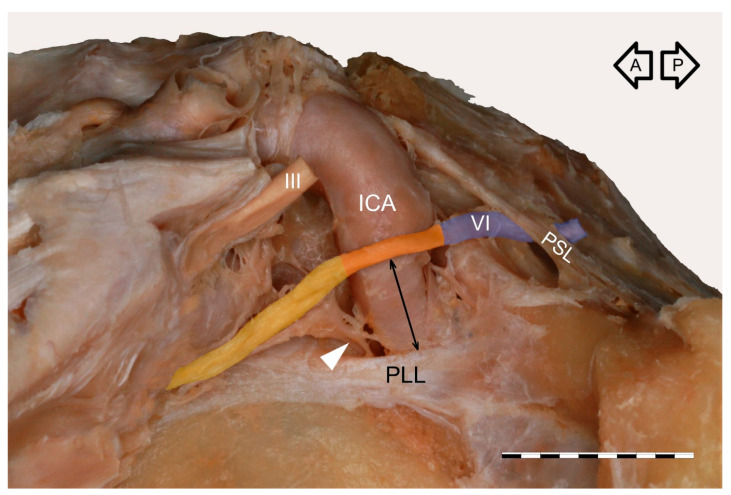
Proposal for subdivision of AN cavernous segment into the carotid portion (marked by orange) located at the point of contact with the cavernous part of the ICA and prefissural portion (pale yellow) located between the anterior aspect of the vertical part of the cavernous part of the ICA and the superior orbital fissure. Wet specimen of the left CS. The distance between the carotid portion of the AN and the petrolingual ligament (PLL) is marked by a double arrow line. The gulfar segment of the AN is colored blue. The white arrowhead marks an anterior bundle of the sympathetic nerve fibers joining the AN’s inferior aspect. III—oculomotor nerve; PSL—petrosphenoidal ligament. The black arrows indicate the anterior (A) and posterior (P) directions. Scale bar corresponds to 10 mm.

**Figure 10 brainsci-11-00649-f010:**
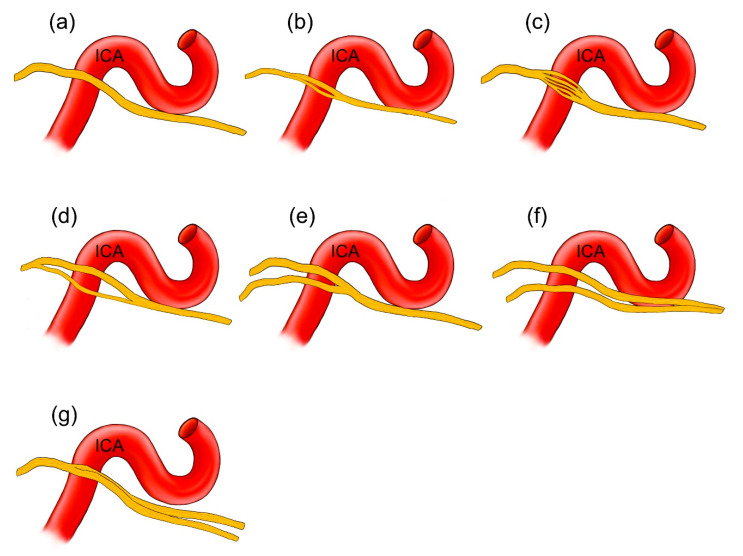
Schematic drawings illustrating anatomical variations of the AN. The nerve is colored yellow. (**a**) The most common variant found was a single nerve trunk. (**b**,**c**) Branching of the nerve into separate rootlets (“pseudobranching”) at the crossing site with the posterior vertical portion of the cavernous part of the ICA. This variant is seen only at the nerve’s cavernous segment. (**d**–**f**) show AN duplication. In (**d**), a single nerve enters the petroclival venous confluence. Variants (**d**,**e**) represent two separate parts of the duplicated AN merging within the CS. In another anatomical variant (**f**), two nerve trunks travel the entire intracranial course of the AN. (**g**) Early division of the AN into two branches before entering the orbit.

**Table 1 brainsci-11-00649-t001:** The frequency of the AN anatomical variations found in the examined sample.

Specific Anatomical Variant	Number of Cases	Frequency Observed
Typical AN with a single trunk	38/60	63.3%
Short division of the cavernous portion of the AN into separate rootlets (“pseudobranching”)	15/60	25%
Duplication of the AN	4/60	6.7%
Early division of the AN (before reaching the orbit) into separate branches	3/60	5%

## Data Availability

Data are contained within the article.

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
