# Peer review of "Gross and Micro-Anatomical Study of the Cavernous Segment of the Abducens Nerve and Its Relationships to Internal Carotid Plexus: Application to Skull Base Surgery"

_brainsci, 2021, doi:10.3390/brainsci11050649_

Round 1

Reviewer 1 Report

Dear editors, dear authors,

thank you for the nice manuscript, which is thoroughly elaborated, though the presented findings remain on a descriptive level. I have no further suggestions.

Thus, in my opinion it is rather a decision of the editorial board if the manuscript lies withinthe scope of the journal.

Author Response

Dear editors, dear authors,

thank you for the nice manuscript, which is thoroughly elaborated, though the presented findings remain on a descriptive level. I have no further suggestions.

Thus, in my opinion it is rather a decision of the editorial board if the manuscript lies within the scope of the journal.

Answer: We thank the reviewer for positive comments on our manuscript.

Reviewer 2 Report

The manuscript “Gross and micro-anatomical study of the cavernous segment of the abducens nerve and its relationships to internal carotid plexus: Application to skull base surgery” by Grzegorz Wysiadecki, Maciej Radek, R. Shane Tubbs, Joe Iwanaga, Jerzy Walocha, Piotr BrzeziÅ„ski and MichaÅ‚ Polguj examine both the macroscopic and microscopic anatomical variations of the cavernous segment of the abducens.

The authors expand their own previous investigations on the anatomical variations in the intracranial course of the abducens nerve - Wysiadecki et al. (2015) The abducens nerve: its topography and anatomical variations in intracranial course with clinical commentary. Folia Morphol 74:236-244 – deepening in the statistical evaluation of the anatomical variations and in the relationship between the abducens nerve and the internal carotid plexus. Moreover, they apply to the new study their expertise with Sihler’s staining and introduce a revealing histological study.

The group's extensive experience in this field of study is a guarantee of the quality of the work. Actually, the standard of the manuscript is very high. The information is very clearly exposed, well organised and enlightening. The writing is proper, and the discussion is clear and very comprehensive. The methods are described carefully and in a detailed way. The quality of the pictures and their labelling is excellent.

The results of the macroscopic study are perfectly linked to the microscopic study, allowing the reader to establish an easy correlation between both approaches, and finally the nice schematic drawing of the variations would be very helpful for the reader.

This is undoubtedly a magnificent piece of neuroanatomical research that will be useful both in the performance of surgery and in imaging studies involving the cavernous sinus.

Minor issues:

  • It would be interesting for the authors to include in the results section a table summarising the frequency of the variations found.
  • The micrographs, figures 4 to 8, should compulsorily include a scale bar. The indication of the lens used is not relevant as the magnification of the photo also depends on the magnification of the eyepiece, adapters, not to mention possible digital processing when editing the manuscript (zoom in and out, cropping, resizing, ...). Only a scale bar allows to know the actual measure of the features pictured.
  • In this paragraph of the discussion:

“Such a term seems to be justified as the individual fascicles merge into a common trunk after a short course ranging from 7.1 mm to 12.9 mm  in this study, from 5 mm to 12 mm according to Zhang et al. [20], and from 4.5 mm to 9.5 mm (mean: 8.4 mm, SD: 1.8 mm) according to another report [25].

The authors should indicate that reference 25 corresponds to their previous study. It would therefore be coherent for them to explain succinctly the reasons for the quantitative differences they found between the two studies.

  • The anatomical study of the sympathetic pathways in the sinus cavernous has always been a subject of great importance. For this reason, the study of the anastomoses of sympathetic fibers with the branches of the abducens nerve is of special interest. In the absence of immunohistochemistry, for example with anti-tyrosine-hydroxylase, which would contrast the sympathetic fibers, it would be useful to have images at higher magnification to demonstrate in transverse sections of the abducens nerve its myelinic character, as opposed to the amyelinic character of the sympathetic fibers.

Author Response

This is undoubtedly a magnificent piece of neuroanatomical research that will be useful both in the performance of surgery and in imaging studies involving the cavernous sinus.

Answer: We thank the reviewer for their thoughtful and thorough review, which helped improve the manuscript.

It would be interesting for the authors to include in the results section a table summarising the frequency of the variations found.

Answer: Thank you for your suggestion. We added Table No. 1, summarising the frequency of the variations found in our study. This table was included in the results section.

The micrographs, figures 4 to 8, should compulsorily include a scale bar. The indication of the lens used is not relevant as the magnification of the photo also depends on the magnification of the eyepiece, adapters, not to mention possible digital processing when editing the manuscript (zoom in and out, cropping, resizing, ...). Only a scale bar allows to know the actual measure of the features pictured.

Answer: We thank the reviewer for this crucial remark. Scale bar corresponding to 1 mm was added to Figures 4 to 8.

In this paragraph of the discussion: “Such a term seems to be justified as the individual fascicles merge into a common trunk after a short course ranging from 7.1 mm to 12.9 mm in this study, from 5 mm to 12 mm according to Zhang et al. [20], and from 4.5 mm to 9.5 mm (mean: 8.4 mm, SD: 1.8 mm) according to another report [25]. The authors should indicate that reference 25 corresponds to their previous study. It would therefore be coherent for them to explain succinctly the reasons for the quantitative differences they found between the two studies.

Answer: We introduced clarification: "according to another report of Wysiadecki et al. [25]," strictly pointing to the previous report. We want to stress that the present study was conducted on an entirely new sample. It can be the first reason for the quantitative differences that we found between the two studies. However, anatomical variability seems to be the fundamental reason for differences in the distance on which pseudobranching of the abducens nerve was observed. Pseudobranching (division into a few separate rootlets) can involve the cavernous segment of the nerve at a very variable distance. Thus, we added the following explanation to the discussion: "This indicates that pseudobranching of the AN can take place in various ways and at a different distance; however, when present, pseudobranching always involves the nerve's point of contact with the posterior vertical segment of the cavernous ICA."

The anatomical study of the sympathetic pathways in the sinus cavernous has always been a subject of great importance. For this reason, the study of the anastomoses of sympathetic fibers with the branches of the abducens nerve is of special interest. In the absence of immunohistochemistry, for example with anti-tyrosine-hydroxylase, which would contrast the sympathetic fibers, it would be useful to have images at higher magnification to demonstrate in transverse sections of the abducens nerve its myelinic character, as opposed to the amyelinic character of the sympathetic fibers.

Answer: We thank the reviewer for this suggestion. We added a new Figure 5C. We demonstrated higher magnification (under 10x objective) of a transverse section of the abducens nerve. For this purpose, we used specimen stained with the Gordon and Sweets technique since this method is characterized by satisfactory contrast. In Figure 5C, the visual difference in the content of fibers having myelin sheath can be observed between abducens nerve fascicles and those derived from the internal carotid plexus. However, it must be stressed that this assessment is visual only. Further immunohistochemical studies are necessary to trace the distribution of sympathetic fibers within the cavernous sinus and in the cerebral arteries' course. We described this limitation at the end of the discussion.

Reviewer 3 Report

This is an elegant and precious anatomic study about the cavernous segment of the abducens nerve. The microscopy and histology are exhaustive and the proposal for the refinement of its classification in segments is supported by the results.

I congratulate the authors for the brilliant pictures.

Author Response

This is an elegant and precious anatomic study about the cavernous segment of the abducens nerve. The microscopy and histology are exhaustive and the proposal for the refinement of its classification in segments is supported by the results.

I congratulate the authors for the brilliant pictures.

Answer: We thank the reviewer for their thoughtful review of our work and kind words.